# 4-1BB-Based CAR T Cells Effectively Reverse Exhaustion and Enhance the Anti-Tumor Immune Response through Autocrine PD-L1 scFv Antibody

**DOI:** 10.3390/ijms24044197

**Published:** 2023-02-20

**Authors:** Kang Cheng, Xiangming Feng, Zhirong Chai, Zhenzhen Wang, Zheng Liu, Zhanchao Yan, Yanming Wang, Shoutao Zhang

**Affiliations:** 1Laboratory of Biotechnology Drugs, School of Life Sciences, Zhengzhou University, Zhengzhou 450001, China; 2Laboratory of Epigenetics and Translational Medicine, School of Life Sciences, Henan University, Kaifeng 475004, China; 3Longhu Laboratory of Advanced Immunology, Zhengzhou 450001, China

**Keywords:** PD-L1 scFv antibody secreting CAR T, 4-1BB, T cell exhaustion, immune checkpoint

## Abstract

Exhaustion of chimeric antigen receptor (CAR) T cells is one of the limitations for CAR T efficacy in solid tumors and for tumor recurrence after initial CAR T treatment. Tumor treatment with a combination of programmed cell death receptor-1 (PD-1)/programmed cell death ligand-1 (PD-L1) blockage and CD28-based CAR T cells has been intensively studied. However, it remains largely unclear whether autocrine single-chain variable fragments (scFv) PD-L1 antibody can improve 4-1BB-based CAR T cell anti-tumor activity and revert CAR T cell exhaustion. Here, we studied T cells engineered with autocrine PD-L1 scFv and 4-1BB-containing CAR. The antitumor activity and exhaustion of CAR T cells were investigated in vitro and in a xenograft cancer model using NCG mice. CAR T cells with autocrine PD-L1 scFv antibody demonstrate enhanced anti-tumor activity in solid tumors and hematologic malignancies by blocking the PD-1/PD-L1 signaling. Importantly, we found that CAR T exhaustion was largely diminished by autocrine PD-L1 scFv antibody in vivo. As such, 4-1BB CAR T with autocrine PD-L1 scFv antibody combined the power of CAR T cells and the immune checkpoint inhibitor, thereby increasing the anti-tumor immune function and CAR T persistence, providing a cell therapy solution for a better clinical outcome.

## 1. Introduction

Chimeric antigen receptor (CAR) T cells are T cells engineered with a designed synthetic receptor, which targets specific antigens of tumor cells and activates T cells without the aid of MHC molecules [1,2]. Anti-CD19 CAR T cells are already used for the treatment of leukemia with a high success rate [3]. However, nearly half of the patients experience tumor recurrence after anti-CD19 CAR T cells treatment [3], and CAR T cells are much less successful in the treatment of solid tumors [4]. In the tumor microenvironment (TME), CAR T cells are constantly exposed to tumor antigens, soluble immune regulatory factors, hypoxia conditions, and regulatory leukocytes including macrophages and dendritic cells [5,6]. These TME signals collectively lead to CAR T cell exhaustion, including a gradual decrease in the anti-tumor activity, the decline of T cell proliferation, and an increase in T cell death [5,6]. Therefore, it is of high interest to avoid CAR T cell exhaustion during cancer treatment.

A large body of research has revealed that the exhaustion of CAR T cells can be reversed by blocking the PD-1/PD-L1 signaling pathway [7,8]. A combination of CAR T cells with immune checkpoint blocking (ICB) drugs has been used in the treatment of multiple tumors with much improved efficacy [9,10]. In particular, PD-1/PDL1 antibodies applied in combination with CAR T cells are potent to overcome the immunosuppression of TME [7,8]. In a glioma mouse model, PD-1 antibody injection enhanced the anti-tumor effect of CAR T cells and protected CAR T cells from exhaustion and cell death induced by prolonged antigen stimulation [11].

Systemic treatment regimen with PD-1/PDL1 antibody injection may induce immune-related adverse effects (irAEs) that bring patients great pain and even risk of life [12]. Alternatively, the disruption of PD-1 gene in CAR T cells by shRNAs or the CRISPR-Cas9 technique eliminates the functions of PD-1 [13,14]. In mouse xenograft models of glioma, pleural mesothelioma, and ovarian and colorectal cancers, the tumor load was reduced and the effector function of CAR T cells was much improved after PD-1 gene disruption [13,14]. However, these PD-1 mutant CAR T cells cannot impact the TME effects on immune suppression and exhaustion of endogenous immune cells [15]. To take advantage of both CAR T cells and ICB antibodies, CAR T cells were engineered to secrete the PD-1 antibody, which can enhance anti-tumor activity of T cells by autocrine and paracrine pathways [16]. Further, with the tumor homing effects of CAR T cells, autocrine PD-1/PDL1 antibodies show local effects and reduce the risk of irAEs [16].

In the TME, dendritic cells (DCs), tumor-associated macrophages (TAMs), and other immune cells upregulate the expression of PD-L1 [17,18]. In contrast, PD-1 mainly expresses on T cells [17,18]. Therefore, PD-L1 antibody exerts its functions through immune cells different from that of the PD-1 antibody, i.e., blocking PD-L1 not only reduces the inhibition of T cells, but also elicits an anti-tumor response through other immune cells. In a melanoma mouse model, and the PD-L1 antibody prolonged the survival time of tumor-bearing mice more effectively than the PD-1 antibody [19]. Currently, the mechanisms by which PD-L1 antibody reverts T cell exhaustion is largely unknown and needs to be further studied.

Most studies on the reversal of T cell exhaustion by PD-1/PD-L1 blockade focused on CD28-based CAR T cells. 4-1BB-based CAR T cells showed more persistence in vivo compared with CD28-based CAR T [7]. However, whether 4-1BB-based CAR T cells could benefit from self-secreted PD-L1 scFv antibody to reverses T cell exhaustion has not been investigated.

In this study, we designed Her2.BBz.PD-L1 CAR T cells and CD19.BBz.PD-L1 CAR T cells for the treatment of breast cancer and B-ALL, respectively. Our results show that autocrine PD-L1 scFv antibody reverses T cell exhaustion and improves anti-tumor efficacy, thereby providing a cell therapy solution with potential for a better clinical outcome.

## 2. Results

### 2.1. Co-Expression of PD-L1 scFv Antibody with CARs in Human T Cells

To enhance the antitumor activity of CAR T cells against CD19 positive B-ALL and Her2 positive breast cancers, we inserted CD19.BBz, CD19.BBz.PD-L1, Her2.BBz, and Her2.BBz.PD-L1 CARs into the multi-cloning site of pCDH-EF1α-MCS-CMV-copGFP vector (Figure 1A). We chose the 4-1BB signaling fragment due to its greater persistence of CAR T cells in vivo [7,20]. The IRES (internal ribosome entry site) allows the translation of scFv PD-L1 antibody. Sanger sequencing was performed to verify the constructs’ sequences accuracy.

CD3-positive T cells were isolated with magnetic beads from healthy donor peripheral blood with signed consent. The recombinant lentiviruses with the CAR constructs were packaged and purified to appropriate concentrations. Flow cytometry analyses of the CAR signals found that the transduction efficiency of recombinant lentivirus ranged from 64.44% to 79.86% (Figure 1B), suggesting that CAR-fusion protein can express efficiently in the effector cells after recombinant lentivirus infection. We found that cells with high expression levels of GFP showed high expression levels of Her2-CAR or CD19-CAR (Appendix A), suggesting that the expression of GFP is proportional to the expression of the CAR. Immunofluorescence analyses found that PD-L1 scFv antibody secreted into the culture supernatants from CD19.BBz.PD-L1 and Her2.BBz.PD-L1 CAR T cells recognized PD-L1 on the surface of target cells (Figure 1C). To indirectly quantify PD-L1 scFv section, we tested if PD-L1 scFv antibody can compete with PD-L1 antibody for cancer cell binding in flow cytometry analyses. We found the fluorescence intensity of PD-L1 staining was significantly decreased after incubation with the supernatant of the CD19.BBz.PD-L1 CAR T and Her2.BBz.PD-L1 CAR T cells (Figure 1D), supporting the secretion of PD-L1 scFv antibody to the supernatant.

### 2.2. Autocrine scFv PD-L1 Antibody Promotes the Functions of CD19.BBz CAR T and CD19.BBz.PD-L1 CAR T Cells against Nalm6-Luc Cells

To facilitate the analyses of the anti-tumor activity of anti-CD19 CAR T cells in hematologic neoplasms, Nalm6 cells were transduced with a lentivirus vector to express GFP and luciferase. After viral transduction and puromycin selection, the expression of GFP was observed under fluorescence microscope (Appendix A). Flow cytometry analyses found 97.9% of Nalm6 cells were GFP-positive cells (Appendix A).

The cytotoxicity of anti-CD19 CAR T cells was measured by the luciferase activity as an indicator of Nalm6-Luc cell viability. We found that the cytotoxicity of CD19.BBz.PD-L1 CAR T cells was significantly stronger than that of CD19.BBz CAR T cells at 5:1 and 10:1 effector:target ratios after 48 h coculture (Figure 2A, top panel). IL-2 and IFN-γ cytokines are important regulators for the anti-tumor activity of CAR T cells. We found that IL-2 (Figure 2A, middle panel) and IFN-γ (Figure 2A, bottom panel) produced by CD19.BBz.PD-L1 CAR T cells were significantly higher than that produced by CD19.BBz CAR T cells at 72 h of effector and target cells coincubation. These results indicate PD-L1 scFv antibody autocrine anti-CD19 CAR T cells possess stronger anti-tumor activity against Nalm6-Luc cells.

### 2.3. Autocrine scFv PD-L1 Antibody Promotes the Functions of Her2.BBz CAR T and Her2.BBz.PD-L1 CAR T Cells against HCC1954-Luc Cells

To facilitate the analyses of the anti-tumor activity of anti-Her2 CAR T cells in solid tumors, we chose breast cancer HCC1954-Luc cells expressing a luciferase reporter gene. The HCC1954-Luc cells was obtained by the same methods of Nalm6-Luc (Appendix A). Using the luciferase activity as an indicator of cell viability, we found the cytotoxicity of Her2.BBz.PD-L1 CAR T cells was significantly higher than that of the Her2.BBz CAR-T cells at effector–target ratios of 10:1 (Figure 2B, top panel). Next, we examined the cytokine secretion of CAR T cells by ELISA; we found that IL-2 (Figure 2B, middle panel) and IFN-γ (Figure 2B, bottom panel) secreted by Her2.BBz.PD-L1 CAR T cells were significantly higher than that produced by Her2.BBz CAR T cells at effector–target ratios of 10:1 at 72 h of effector and target cells coincubation. Taken together, the above results indicate that PD-L1 scFv antibody autocrine anti-Her2 CAR T cells possess stronger anti-tumor activity against HCC1954-Luc cells.

### 2.4. ScFv PD-L1 Antibody Reverses Exhaustion Phenotypes of CD19.BBz CAR T and CD19.BBz.PD-L1 CAR T Cells after Constant Antigen Exposure In Vitro

To assess whether CD19 CAR T cells with autocrine PD-L1 scFv antibody can affect exhaustion, we applied a constant antigen exposure protocol illustrated in Figure 3A. The proliferation of blank T cells and CD19.BBz CAR T cells significantly decreased compared with that of CD19.BBz.PD-L1 CAR T cells (Figure 3B), indicating that autocrine PD-L1 scFv antibody promotes CAR T cell division. Moreover, we found that CD19.BBz CAR T cells showed significantly less cytotoxicity and IL-2 or IFN-γ secretion after constant antigen exposure for 9 days (Figure 3C). In contrast, the cytotoxicity and IL-2 or IFN-γ secretion of CD19.BBz.PD-L1 CAR T cells were preserved after constant antigen exposure (Figure 3C). Interestingly, the expression of immune-suppressive T cell receptors PD-1, TIM-3 (T cell immunoglobulin and mucin domain-containing protein 3), and CTLA-4 (cytotoxic T-lymphocyte-associated protein 4) was comparable between CD19.BBz and CD19.BBz.PD-L1 CAR T cells (Figure 3D), indicating the constant antigen exposure along in vitro do not elicit a significant difference in PD-1, TIM-3 and CTLA-4 expression (Figure 3D).

To test if the secreted scFv PD-L1 antibody affects CAR T cell functions after constant antigen exposure, culture supernatants from CD19.BBz and CD19.BBz.PD-L1 CAR T cells were used to treat CD19.BBz CAR T cells. Supernatant with scFv PD-L1 antibody significantly increased the cytotoxicity of and the secretion of IL-2 and IFN-γ by CD19.BBz CAR T cells that experienced constant antigen exposure for 9 days. (Figure 3E), suggesting that the scFv PD-L1 antibody can have a beneficial impact on CAR T cell functions through paracrine.

### 2.5. ScFv PD-L1 Antibody Reverses Exhaustion Phenotypes of Her2.BBz CAR T and Her2.BBz.PD-L1 CAR T Cells after Constant Antigen Exposure In Vitro

To study Her2.BBz and Her2.BBz.PD-L1 CAR T cell exhaustion after constant antigen exposure, we used the experimental scheme illustrated in Figure 4A. After prolonged exposure to HCC1954-Luc breast cancer cells, the proliferation ability of Her2.BBz CAR T cells was significantly decreased compared with that of Her2.BBz.PDL1 CAR T cells (Figure 4B). Similarly, after prolonged culture with HCC1954-Luc cells, Her2.BBz CAR T cells showed less cytotoxicity and less IL-2 and IFN-γ secretion than Her2.BBz.PD-L1 CAR T cells (Figure 4C), suggesting that autocrine PD-L1 promotes T cell proliferation, antitumor activity, and cytokine secretion. Moreover, the expression of PD-1, CTLA-4, and TIM-3 do not change significantly between Her2.BBz CAR T and Her2.BBz.PD-L1 CAR T cells (Figure 4D), suggesting that prolonged antigen simulation did not significantly alter the expression of these cell markers in vitro. In addition, the cytotoxicity and the cytokine secretion of Her2.BBz CAR T cells were significantly increased upon treatment with the culture supernatant containing the scFv PD-L1 antibody, suggesting that PD-L1 antibody can exert a paracrine effect.

### 2.6. ScFv PD-L1 Antibody Promotes Anti-Tumor Efficacy and Attenuates CAR T Cell Exhaustion in Nalm6-Luc Xenograft Tumors In Vivo

T cells from three healthy donors were used in in vivo experiments, and the mice experiment was performed once for each donor. The results obtained are from one donor and are representative of three donors. To assess the anti-tumor efficacy of CD19.BBz and CD19.BBz.PD-L1 CAR T cells, we applied an experimental procedure as shown in Figure 5A. Compared with the CD19.BBz CAR T cell treatment group, the tumor burden in the CD19.BBz.PD-L1 CAR T cell treatment group significantly decreased, the luciferase activity reduced approximately 10-fold on day 7 and 100-fold on day 14 (Figure 5B). In contrast, the tumor burden of CD19.BBz CAR T group increased from day 7 to day 14, indicating that the CD19.BBz CAR T cells are not effective in preventing leukemia progression (Figure 5B). The body weight of CD19.BBz.PD-L1 CAR T group were relatively stable, but the body weight of CD19.BBz CAR T group start decreased sharply on day 17 (Figure 5C). Moreover, on day 27 after treatment, 100% survival was observed for the CD19.BBz.PD-L1 CAR T cell treatment group (Figure 5D), while only 40% survival for the CD19.BBz CAR T cell treatment group was observed (Figure 5D). Above results indicate that CD19.BBz.PD-L1 CAR T cells exert a strong and long-lasting anti-tumor effects. To evaluate the T cell exhaustion in vivo, T cells were isolated from mouse blood at day 14 after treatment. Interestingly, the number of CD4^+^ CAR-T cells and CD8^+^ CAR-T cells were significantly increased in the group with CD19.BBz.PD-L1 CAR T cell treatment than the group with CD19.BBz CAR T cell treatment (Figure 5E). Moreover, the expression of T cell exhaustion markers including PD-1, CTLA-4, and TIM-3 in the CD19.BBz.PD-L1 CAR T cell treatment group was significantly decreased, suggesting that scFv PD-L1 antibody can promote T cell functions and attenuate exhaustion via its immunomodulatory effects in vivo (Figure 5F).

### 2.7. ScFv PD-L1 Antibody Promotes Anti-Tumor Efficacy and Attenuates Exhaustion of CAR T Cells in HCC1954 Breast Cancer Xenograft Tumors In Vivo

T cells from three healthy donors were used in in vivo experiments, and the mice experiment was performed once for each donor. The results obtained are from one donor and are representative of three donors. To further evaluate the therapeutic efficacy of scFv autocrine PD-L1 antibody CAR T cells in solid tumors, we applied an experimental scheme as illustrated in Figure 6A. Similarly, the tumor growth curve (Figure 6B), and the weight and size of dissected tumors (Figure 6C,D) of Her2.BBz.PD-L1 CAR T cell treatment group were significantly decreased compared with those of the Her2.BBz CAR T cell treatment group. As such, autocrine scFv PD-L1 antibody promotes the antitumor efficacy of anti-Her2 CAR T cells.

To evaluate the exhaustion of CAR T cells, tumor infiltrating T cells were isolated from the HCC1954 xenograft tumors. We found that the number of CD4^+^ CAR-T cells and CD8^+^ CAR-T cells were significantly increased in the group with Her2.BBz.PD-L1 CAR T cell treatment than the group with Her2.BBz CAR T cell treatment (Figure 6E). Moreover, compared with the Her2.BBz CAR T cell treatment group, the expression of PD-1, CTLA-4 and TIM-3 were lower in the Her2.BBz.PD-L1 CAR T cell treatment group (Figure 6F). Therefore, the autocrine PD-L1 scFv antibody reduced the expression of the T cell exhaustion regulatory factors in the TME of solid tumors.

## 3. Discussion

The Food and Drug Administration (FDA) has approved five products of CAR T for clinical application. Kymriah, the first CAR T product, has a sustained release rate of 81% with CD19 special CAR T cells [21]. However, there is still a considerable number of patients with limited or no clinical response [21]. Because of the immunosuppression effects of TME and cell type diversity, the antitumor ability of CAR T cells in solid tumors is further restricted [22]. Moreover, the constant stimulation CAR T cells by cancer antigens results in T cell exhaustion and limited therapeutic efficacy [5]. As such, technologies to attenuate T cell exhaustion and decrease immunosuppression bear the promise for effective CAR T cell treatment to improve anti-tumor activity and prevent tumor relapse.

The PD-1/PD-L1 pathway downregulates anti-tumor immune response, inhibits T cell cytotoxicity in the TME, and mediated T cell exhaustion [23]. In addition, compared with LAG-3 and TIM-3, PD-1 plays a more important role in mediating T cell exhaustion [8]. As such, the combination of CAR T and PD-1/PD-L1 pathway inhibitors can achieve better T cell-mediated antitumor effect.

Multiple techniques have been developed to block PD-1/PD-L1. CAR T cells targeting PSMA (prostate-specific membrane antigen) can enhance the anti-tumor activity and prolong the survival time of mice in prostate cancer model by combining with the PD-1 antibody [24]. However, PD-1/PD-L1 blockers not only cause irAE, but also require repeated dosing to reach an effect. Moreover, PD-1/PD-L1 antibodies can be captured by macrophages before exerting its effects on T cell activation, leading to drug insensitivity and tumor resistance [25,26]. The extracellular domain of PD-1 is used as the PD-L1 binding domain of CAR T to construct a dominant negative receptor, which reduced T cell exhaustion, increased cytotoxicity and factor secretion of CAR T cells [27]. The CAR T cells with PD-1 encoding gene PDCD1 (programmed cell death 1) silencing do not affect the immune response of endogenous immune cells [15]. Therefore, autocrine PD-1/PD-L1 antibody from CAR T cells has additional benefits to reach optimized anti-tumor activity compared with the above three techniques.

CAR T cells were previously engineered to secrete the PD-1 scFv antibody, which enhanced the anti-tumor activity in the syngeneic and xenogeneic mouse models and prolonged the survival time of mice [16]. In addition, the scFv antibody concentration in circulation was very low, thereby reducing systemic side effects [16]. The blocking of PD-1 and PD-L1 often have different immunological effects, while the blocking of PD-L1 not only activates T cells, but also elicits a non-T-cell (e.g., TAMs)-dependent anti-tumor response. However, it is unknown whether autocrine PD-L1 scFv antibody can attenuate CAR T cell exhaustion in the TME while promote the anti-tumor activity of CAR T cells.

In our study, we selected Her2 and CD19 as targets for B-ALL and breast cancers, respectively. We designed an autocrine scFv PD-L1 antibody to study if binding of scFv PD-L1 antibody to the PD-L1 positive Nalm6 B-ALL and HCC1954 breast cancer cells can affect CAR T cell functions. In vitro cytotoxicity tests showed that compared with traditional CAR T cells, CAR T cells engineered with autocrine scFv PD-L1 antibody had higher cytotoxicity and IL-2 and IFN-γ secretion. Moreover, upon being transplanted into mice with allogeneic B-ALL developed from the Nalm6-Luc cells, CD19.BBz.PD-L1 CAR T cells were much more efficient in cancer inhibition than the CD19.BBz CAR T cells (Figure 5). The tumor load in the CD19.BBz.PD-L1 CAR T cell treatment group was significantly decreased by 10–100-fold over time and prolonged mouse survival. Moreover, we found that the autocrine scFv PD-L1 antibody enhanced the antitumor efficacy of Her2.BBz CAR T cells against HCC1954 breast cancer xenograft tumors, showing a significant decrease in tumor growth and tumor weight (Figure 6). Taken together, the above results indicate that autocrine scFv PD-L1 antibody CAR T cells can enhance the anti-tumor activity.

The mechanism of T cell exhaustion remains an area of study of high interest. Here, we further studied the effects of autocrine scFv PD-L1 antibody on CAR T cell exhaustion. PD-1 binding to PD-L1 leads to the dephosphorylation of CD28 thereby, inhibiting T cell activation and inducing T cell exhaustion [8]. However, even in the presence of the PD-1 antibody, CD28-based CAR T cell exhaustion can still occur, indicating that there are other exhaustion mechanisms. 4-1BB-based CAR T cells show more persistent activation in vivo than CD28 based CAR T cells by eliciting a different cell signaling cascade [8,28]. A recent clinical trial found that 12 patients receiving CD19.BBz CAR T cell treatment had poor antitumor effect. The PD-1 antibody Pembrolizumab was used for follow-up treatment, and four patients were in remission [29]. The activation and proliferation of CAR T cells were increased, and T cell exhaustion was decreased in clinical responders [29]. Currently, most studies on the reversal of T cell exhaustion by PD-1/PD-L1 blockade focus on CD28-based CAR T. In our studies, we further tested whether autocrine scFv PD-L1 antibody can affect 4-1BB-based CAR T exhaustion in vitro and in vivo. In the experiment of anti-Her2 CAR T cell dysfunction (Figure 4C), we found the cyto-toxicity of Her2.BBz has no significant differences from blank-T on day 0 (donor 4), the proportions of CAR+ CD4+ and CAR+ CD8+ T cells were measured by flow cytometry and the result showed proportionately more CD4+ cells than CD8+ cells (Appendix A, left panel). We observed the ratio of CD4+:CD8+ was close to 1:1 in the CAR T cell population from another donor with the similar transduction efficiency (Appendix A, right panel). We found that the cytotoxicity of CAR T from donor 2 was significantly higher than donor 4, and the secretion of the IL-2 and IFN-γ from donor 2 CAR T cells significantly decreased compared with donor 4 at the E:T ratio of 10:1 (Appendix A). As such, we speculate that the cytotoxicity and cytokine secretion may be related to the ratio of CD4+ T: CD8+ T cells in the CAR T population. After prolonged cancer cell antigen exposure, autocrine scFv PD-L1 antibody promoted the proliferation ability, cytotoxicity, and IL-2 and IFN-γ secretion of CD19.BBz.PD-L1 and Her2.BBz.PD-L1 CAR T cells, suggesting that autocrine scFv PD-L1 antibody can improve T cell functions. Moreover, the culture supernatant containing the scFv PD-L1 antibody could enhance the cytotoxicity and the cytokine secretion of CD19.BBz and Her2.BBz CAR T cells, suggesting that scFv PD-L1 antibody can serve as a paracrine signal to improve T cell functions. Furthermore, scFv PD-L1 antibody decreased the expression of T cell exhaustion markers PD-1, TIM-3, and CTLA-4 in xenograft tumor models in vivo but not in cultured cells in vitro, suggesting that additional mechanisms are underlying the regulation of PD-1, TIM-3, and CTLA-4 expression in the TME.

As such, the mechanism of T cell exhaustion remains to be further studied. PD-1/PD-L1 antibodies can prevent T cell exhaustion by preventing CD28-mediated CAR T exhaustion. In contrast, blocking PD-1/PD-L1 does not directly regulate 4-1BB, which may explain why the T exhaustion-related markers did not change significantly in the in vitro exhaustion analyses. Since PD-L1 is also expressed in immunosuppressive cells in the TME, and T cell exhaustion is regulated by these immune cells in vivo, it may explain why scFv PD-L1 antibody may affect PD-1, TIM-3 and CTLA-4 via other cells in the TME. Our future work will analyze why PD-L1 antibody prevents 4-1BB-based CAR T exhaustion.

In summary, we demonstrate that autocrine PD-L1 scFv antibody enhances the anti-tumor effect of CAR T cells and found that 4-1BB-based CAR T cell exhaustion can be prevented by the autocrine PD-L1 scFv antibody. Our work lays a foundation for studying the mechanisms underlying T cell exhaustion and provides an effective CAR T treatment strategy for future clinical studies.

## 4. Materials and Methods

### 4.1. The Cell Lines and Medium

Human breast cancer HCC1954 (CRL-2338, ATCC), HCC1954-Luc cells, human acute lymphoid leukemia Nalm6 cells (CRL-3273, ATCC), and Nalm6-Luc cells were cultured in RPMI1640 medium (Gibco, Grand Island, NY, USA) supplemented with 10% fetal bovine serum (Gibco, Grand Island, NY, USA), 100 µg/mL streptomycin, and 100 IU/mL penicillin (Gibco). Human HEK-293T cells (ACS-4500, ATCC) were cultured in Dulbecco’s modified Eagle’s medium (DMEM) (Gibco, Grand Island, NY, USA) supplemented with 10% fetal bovine serum (Gibco), 100 µg/mL streptomycin, and 100 IU/mL penicillin (Gibco, Grand Island, NY, USA). Human primary T cells were cultured in X-VIV0-15 medium (Lonza, Walkersville, MD, USA) containing 5% autologous serum and 40 IU/mL human IL-2 (PeproTech, Rocky Hill, NJ, USA). All cells were cultured in a 5% CO_2_ incubator at 37 °C.

### 4.2. Molecular Cloning

The CAR was designed with the Her2 or CD19 scFv antibody for target binding, the CD8 transmembrane domain, and the co-stimulatory domain of 4-1BB and CD3ζ with or without the scFv PD-L1 antibody. The DNA sequences encoding CARs were synthesized by Sangon Biotech (Sangon Biotech, Shanghai, China). The CAR sequences were cloned into the pCDH lentivirus expression vector and sequence confirmed. The sequences of CD19 scFv, Her2 scFv, and PD-L1 scFv antibody were obtained from Chinese patents No. 201510233748.0, No. 201710013312.X, and No. 201680027181.4, respectively. PD-L1 scFv antibody was downstream of CARs with an IRES sequence for translation initiation (Appendix A). Four CARs were constructed to produce lentivirus, namely CD19.BBz, CD19.BBz.PD-L1, Her2.BBz, and Her2.BBz.PD-L1.

### 4.3. Lentiviral Production

The methods of viral packaging and concentration were performed as previously described [30]. Briefly, the recombinant lentivirus was produced by co-transfecting HEK-293T cells using lipofectamine 3000 with 10 μg various pCDH lentiviral CAR expression vectors or the pCDH-luciferase-P2A-copGFP vector, and the packing plasmids (7.5 μg psPAX2 and 2.5 μg pMD2G) in a 10 cm petri dish. At 24 h and 52 h after transfection, the lentivirus particles were harvested and filtered with 0.45 µm micron filters (Millipore, Billerica, MA, USA). The viral supernatants were harvested and stored at −80 °C.

To determine viral titers, the HEK-293T cells were seeded in 96 well plate at 1 × 10^4^ per well. The next day, the virus was 10-fold serial diluted by complete media and added to 96 well plate. Then, 24 h later, medium was replaced and was cultured for 48 h, then cells were harvested and GFP fluorescence was measured by flow cytometry to calculate viral titers.

### 4.4. Generation of Luciferase Expressing Nalm6-Luc Cells and HCC1954-Luc Cells

Nalm6 cells and HCC1954 cells were transduced at MOI (multiplicity of infection) of 1 in medium supplemented with 5 μg/mL of polybrene (Sigma, Saint Louis, MO, USA, TR-1003) and Nalm6 cells was centrifuged at 1500× *g* for 90 min at 31 °C. After 24 h, fresh medium was added, and cells were cultured for another 48 h. After being selected with 5 ug/mL puromycin (Gibco, Grand Island, NY, USA, A1113803), GFP expression was analyzed by fluorescence microscope and flow cytometry.

### 4.5. Generation of CAR T Cells

Human primary T cells were isolated from the peripheral blood of 4 healthy donors. The study with human samples was approved by the Ethics Committee of Henan University, and informed consent was obtained from all blood donors. PBMCs were separated by Ficoll (General Electric, Uppsala, Sweden, 17144002) gradient centrifugation at 500× *g* for 15 min at 21 °C. CD3+ T cells were separated from PBMCs by the MACS CD3 microbeads (Miltenyi Biotec, Bergisch Gladbach, Germany, 130-097-043). The obtained CD3+ cells were activated and amplified with magnetic beads coated with CD3/CD28 antibodies (Gibco, Carlsbad, CA, USA, 11131D).

After activation and expansion for 1 day, the recombinant lentiviruses were transduced into T cells at MOI of 10 in medium supplemented with 4 μg/mL of polybrene and centrifuged at 1500× *g* for 90 min at 31 °C. The transduction procedure was repeated after 24 h to increase the transduction efficiency. The cells were cultured in a 5% CO_2_ incubator at 37 °C. At 48 h after the initial transduction, the cells were collected and the fresh culture medium was added. Transduction efficacy was analyzed by flow cytometry at 72 h after the initial transduction.

### 4.6. Flow Cytometry

To evaluate whether CAR T cells can secrete PD-L1 scFv antibody and compete with PD-L1 antibody in binding to PD-L1 on the surface of Nalm6 and HCC1954 cancer cells, 1 × 10^5^ tumor cells were collected and resuspended with the supernatant of CD19.BBz.PD-L1, Her2.BBz.PD-L1 CAR T cells, or with fresh T cell culture medium and incubated for 0.5 h at room temperature; cells were collected by centrifugation and resuspended in phosphate buffered saline (PBS). Tumor cells were further incubated with APC conjugated anti-human PD-L1 for 0.5 h at 4 °C (Biolegend, San Diego, CA, USA, 329708), the APC conjugated isotype mAb was used as a control.

The lentivirus transduction efficiency of T cells was analyzed by incubation transfected T cells with the recombinant human CD19-Fc chimera (Biolegend, San Diego, CA, USA, 789002) or the biotinylated recombinant human HER2-Fc chimera (Biolegend, San Diego, CA, USA, 796506) for 0.5 h at 4 °C. Cells were collected by centrifugation and resuspended in PBS, incubated with PerCP/Cyanine5.5 labeled anti-human IgG Fc (Biolegend, San Diego, CA, USA, 410709) or PerCP/Cyanine5.5 labeled streptavidin (eBioscience, Carlsbad, CA, USA, 45-4317-82) for another 0.5 h at 4 °C before flow cytometry analyses.

To analyze the amount of CD4^+^ and CD8^+^ T cells in the CAR T cells, T cells was isolated from peripheral blood or tumors by MACS CD3 microbeads. The number of T cells was counted by hematocytometer. Dead cells were stained with 0.04% trypan blue (Sigma, Saint Louis, MO, USA, T8154). T cells was then incubated with recombinant human CD19-Fc chimera or biotinylated recombinant human HER2-Fc chimera for 0.5 h at 4 °C, cells were collected by centrifugation and resuspended in PBS, further incubated with PerCP/Cyanine 5.5 labeled anti-human IgG Fc (Biolegend, San Diego, CA, USA, 410709) or PerCP/Cyanine 5.5 labeled streptavidin (eBioscience, Carlsbad, CA, USA, 45-4317-82), followed by adding PE-labeled anti-human-CD4 antibody (Biolegend, San Diego, CA, USA, 300508) and APC-labeled anti-human-CD8 antibody (Biolegend, San Diego, CA, USA, 344722). Cells were incubated with above reagents for another 0.5 h at 4 °C before flow cytometry analyses.

To measure the proportions of CD4+ and CD8+ T cells in the CAR T cells, T cells were collected and incubated with biotinylated recombinant human HER2-Fc chimera (Biolegend, San Diego, CA, USA, 796506) for 0.5 h at 4 °C, cells were collected by centrifugation and resuspended in PBS, incubated with PerCP/Cyanine5.5 labeled strep-tavidin (eBioscience, Carlsbad, CA, USA, 45-4317-82), APC-labeled anti-human-CD4 antibody (BD Pharmingen, San Diego, CA, USA, 555349) and PE-labeled anti-human-CD8 antibody (BD Pharmingen, San Diego, CA, USA, 555635) for another 0.5 h at 4 °C.

To detect the exhaustion markers, T cells were collected and resuspended in PBS, incubated with APC-labeled anti-human-PD-1 antibody (Biolegend, San Diego, CA, USA, 329907), PE-labeled anti-human-CTLA-4 antibody (Biolegend, San Diego, CA, USA, 349905), or APC labeled anti-human-TIM-3 antibody (Biolegend, San Diego, CA, USA, 345011) for 0.5 h at 4 °C. Flow cytometry data were analyzed using the FlowJo VX.0.7 software (FlowJo, LLC, Ashland, OR, USA).

### 4.7. Immunofluorescence

Nalm6 cells were added to tissue culture wells with a poly-lysine coated coverslip and centrifuged at 1000 rpm for 5 min. Next, the cells were washed with PBS three times, fixed with 4% paraformaldehyde, and blocked with 2% BSA dissolved in PBS to avoid nonspecific binding. The supernatants of PD-L1 scFv antibody autocrine CAR T cells were added and incubated at 4 °C for overnight. After washing, cells were incubated with PE conjugated anti-His6 rabbit monoclonal antibody (dilution 1:100, Abcam, Cambridge, UK, ab237338) at room temperature for 1 h. After washing, cells were stained with 0.1 mg/mL DAPI (Sigma, Saint Louis, MO, USA, D9542) dissolved in PBS. The stained cells were observed under a fluorescence microscope (Zeiss, Oberkochen, Germany). HCC1954 cells were seeded in 12 well plate one day in advance and immunostaining was performed similarly as above described for the Nalm6 cells. 

### 4.8. Luciferase Assay for Cytotoxicity

The Luciferase assays were used to measure the cytotoxicity of CAR T cells. Nalm6-Luc or HCC1954-Luc cells were seeded in 96-well plates. CAR T cells were added at the different effector:target ratios as indicated in the figures. The blank well without cells and Nalm6-Luc or HCC1954-Luc cells without T cells were used as controls. After coculture for 48 h, the supernatant was removed and replaced with fresh medium containing 150 μg/mL D-Luciferin potassium salt (Perkin Elmer, Waltham, MA, USA, 122799). After incubation for 10 min at 37 °C, luminescence signals were measured by microplate reader (Bio Tek, Winooski, VT, USA). The percentage of viable cells was calculated as:% viability = 100 × (value of test samples − value of blank wells)/(values of Nalm6-Luc or HCC1954-Luc without T cells − value of blank wells).

### 4.9. Cytokine Secretion Analyses

The supernatant of the medium was collected and filtered with a 0.22 µm micron filter. IL-2 and IFN-γ concentrations were detected by Elisa kit specific for IL-2 (Dakewe Biotech, Shenzhen, China, 1110202) and IFN-γ (Dakewe Biotech, Shenzhen, China, 1110002) following manufacturers’ instructions.

### 4.10. T Cell Exhaustion Analyses In Vitro

First, 1 × 10^6^ Nalm6 cells were seeded in 6 well plates and 1 × 10^6^ HCC1954 cells were seeded in the 10 cm dishes. Then, 1 × 10^5^ blank T and CAR T cells were then added to the tumor cells. After 3 days, T cells were isolated by the MACS CD3 microbeads and counted using a hematocytometer. The culture medium (conditioned medium) was collected and filtered with 0.45 µm micron filters. Next, 1 × 10^5^ T cells resuspended in a medium containing equal of amount of conditioned medium and fresh medium, and co-cultured with 1 × 10^6^ cancer cells for another 3 days. This process was repeated one more time at day 6, and the cell exhaustion experiment lasted for 9 days in total.

### 4.11. Mouse Xenograft Model

All procedures involving animals were conducted under the Animal Care Guidelines issued by Henan University. The NCG female mice (age 6–8 weeks, weight 20–22 g) were purchased from GemPharmatech (Nanjing, China, T001475). Mice were kept at 24 °C with a 12 h/12 h light–dark schedule. After 1 week of housing in the local animal facility, 6 × 10^6^ Nalm6-Luc cells were resuspended in 0.15 mL PBS and tail vein injected into mice. After 7 days, 15 tumor-bearing mice were randomly divided into three groups. Then, 6 × 10^6^ blank-T, CD19.BBz CAR T, and CD19.BBz.PD-L1 CAR T cells resuspended in 0.2 mL PBS were tail vein injected. Tumor progression was monitored by BLI using the Xenogen-IVIS Imager (PerkinElmer, Waltham, MA, USA) on day 0, 7, 14, and 18. Isoflurane was used for mouse anesthesia during luciferase imaging analyses, and D-luciferin potassium salt (Perkin Elmer, Waltham, MA, USA, 122799) was injected intraperitoneally at 0.15 mpk (mg per kg of body weight) at 10 min prior to imaging. Mice survival was monitored every day, and mouse weight was measured every three days. At day 14, mouse peripheral blood was collected from the orbital vein. T cells were isolated by MACS CD3 microbeads (Miltenyi Biotec, Bergisch Gladbach, Germany, 130-097-043), and the exhaustion markers PD-1, CTLA-4, and TIM-3 were measured by flow cytometry. All mice were euthanized at day 27 after treatment.

For the breast cancer mouse model, 5 × 10^6^ HCC1954 cells were resuspended in 0.1 mL PBS and mixed with 0.1 mL matrigel (Corning, Bedford, MA, USA, 356234). The cell mixture was injected subcutaneously into the NCG female mice (age 6–8 weeks, weight 20–22 g). At 10 days after xenograft, mice were randomly divided into three groups with 6 mice in each group. 1 × 10^7^ blank-T, Her2.BBz CAR T, and Her2.BBz.PD-L1 CAR T cells were resuspended in 0.2 mL PBS and tail vein injected at day 10 and day 17. In addition, 20,000 U IL-2/mice (PeproTech, Rocky Hill, NJ, USA, AF-200-02) were intraperitoneal injected every two days starting from day 10. During anti-tumor treatment, tumor volume was measured every five days using a digital caliper, and the tumor volume was calculated using the formula tumor volume (mm^3^) = Width^2^ × Length/2. All mice were euthanized on day 30, and tumors were harvested and weighed. The exhaustion markers of TILs were analyzed by flow cytometry.

### 4.12. Statistical Analysis

Statistical analyses were carried out using the GraphPad Prism 8.0.2 software (GraphPad Software, San Diego, CA, USA). A total of four healthy donors were enrolled in our studies. To avoid confounding variables such as transduction efficiencies and donor-health status related variability, we present data from a representative individual. All the experiments were repeated at least three times. The data for each experimental group were compared using one-way ANOVA or unpaired *t* test. A *p* value less than 0.05 was considered significant.

## Figures and Tables

**Figure 1 ijms-24-04197-f001:**
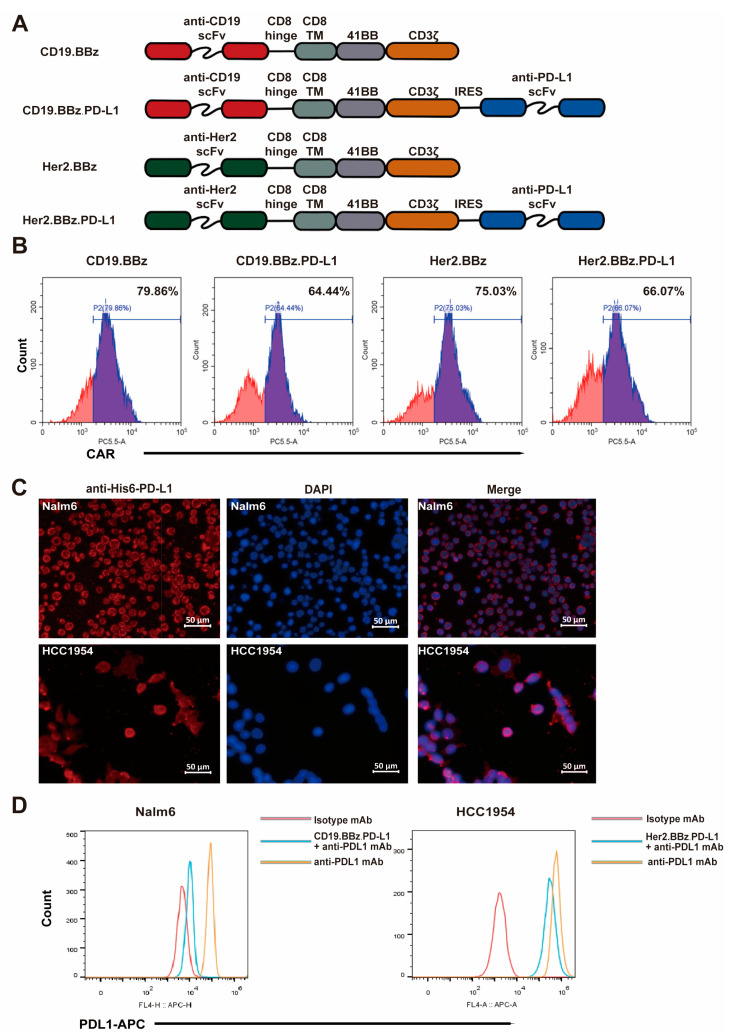
Generation and characterization of chimeric antigen receptor (CAR) T cells. (**A**) Schematic illustration of the CAR constructs designed in this study. (**B**) Flow cytometry analysis of CAR expression in T cells at three days after lentiviral transduction. (**C**) Programmed cell death ligand 1 (PD-L1) expression of Nalm6 and HCC1954 cells were detected with PD-L1 scFv Ab. (**D**) Competition of PD-L1 scFv with the PD-L1 antibody for binding to PD-L1 on Nalm6 and HCC1954 cells was measured by flow cytometry.

**Figure 2 ijms-24-04197-f002:**
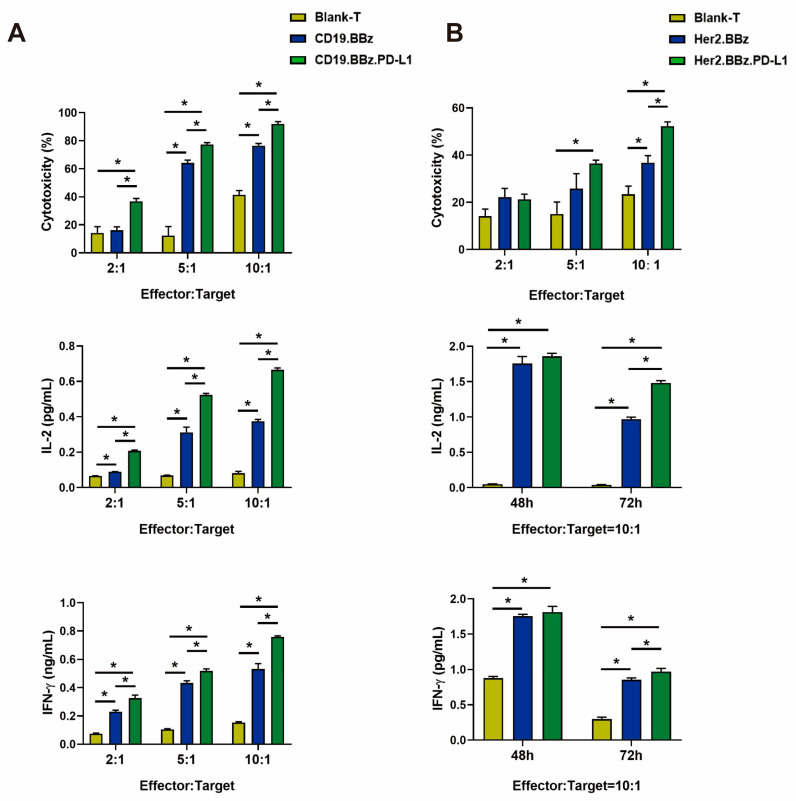
Cancer cell cytotoxicity and cytokine secretion by CAR T cells. (**A**) The cytotoxicity of blank T, CD19.BBz CAR T, and CD19.BBz.PD-L1 CAR T cells against Nalm6-Luc leukemia cells (top panel), and their secretion of IL-2 and IFN-γ cytokines detected by ELISA (middle and bottom panels) (*n* = 3). (**B**) The cytotoxicity of blank T, Her2.BBz CAR T, and Her2.BBz.PD-L1 CAR T cells against HCC1954-Luc breast cancer cells (top panel), and their secretion of IL-2 and IFN-γ cytokines detected by ELISA (middle and bottom panels) (*n* = 3). A total of four healthy donors were enrolled in our studies and we present data from a representative individual. One-way ANOVA was performed to determine statistical significance, the results represent the mean ± SD. (* *p* < 0.05).

**Figure 3 ijms-24-04197-f003:**
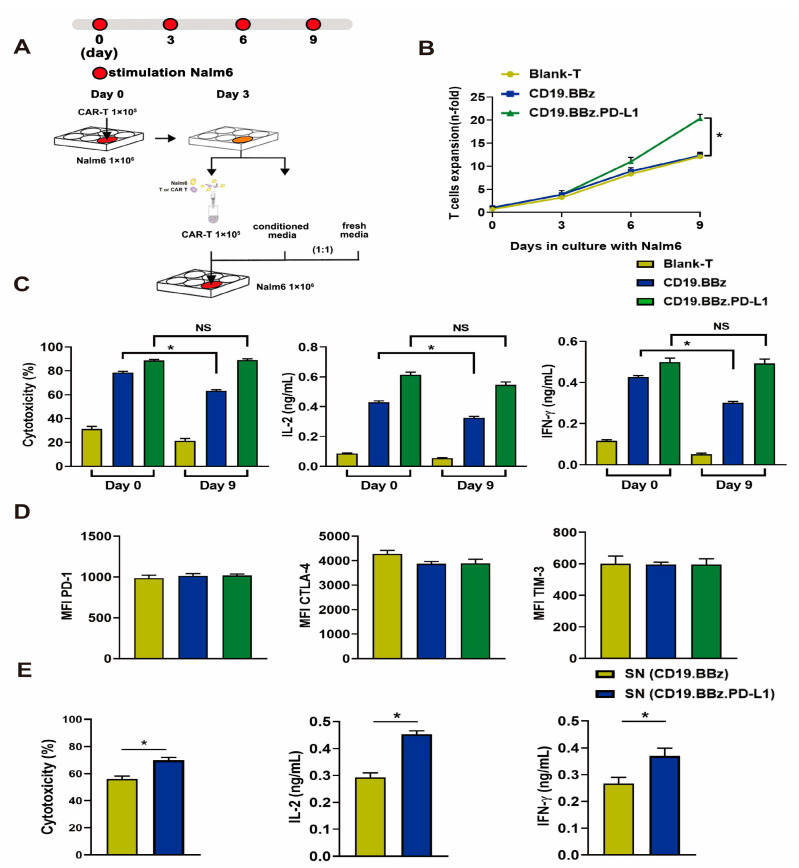
Effects of antigenic stimulation on cell exhaustion, cytotoxicity, cell proliferation, and cytokine secretion by blank T, CD19.BBz CAR T, and CD19.BBz.PD-L1 CAR T cells. (**A**) Scheme of the experimental procedure: 1 × 10^5^ blank T cells and CAR T cells were co-cultured with 1 × 10^6^ Nalm6 cells for 3 days. The culture supernatant (conditioned media) was collected and filtered. T cells were isolated by MACS CD3 microbeads, 1 × 10^5^ T cells resuspended in a medium containing equal amounts of conditioned medium and fresh medium, and co-cultured with 1 × 10^6^ Nalm6 cells for another 3 days. This process was repeated one more time at day 6, and the cell exhaustion experiment lasted for 9 days in total. (**B**) During the prolonged antigenic stimulation, T cells were isolated by CD3 microbeads and analyzed for cell proliferation (*n* = 3). (**C**) The cytotoxicity and the cytokine secretion were measured before (day 0) and after (day 9) antigen stimulation at the E:T ratio of 1:10 (*n* = 3). (**D**) The expression of T cell surface makers PD-1, CTLA-4, TIM-3 was measured by flow cytometry, and the mean fluorescence intensity (MFI) was calculated using FlowJo (*n* = 3). (**E**) The effects of PD-L1 scFv secreted by CAR T cells on the cytotoxicity and cytokine secretion of CD19.BBz CAR T cells. CD19.BBz CAR T cells was co-cultured with Nalm6-Luc cells for 48 h at the E:T ratio of 10:1 (*n* = 3), SN: supernatant. A total of 4 healthy donors were enrolled in our studies and we present data from a representative individual. One-way ANOVA was performed to determine statistical significance (**B**–**D**), *t* test was performed to determine statistical significance (**E**). The results are means ± SD (* *p* < 0.05, compared with CD19.BBz CAR T cells group; NS, no significance).

**Figure 4 ijms-24-04197-f004:**
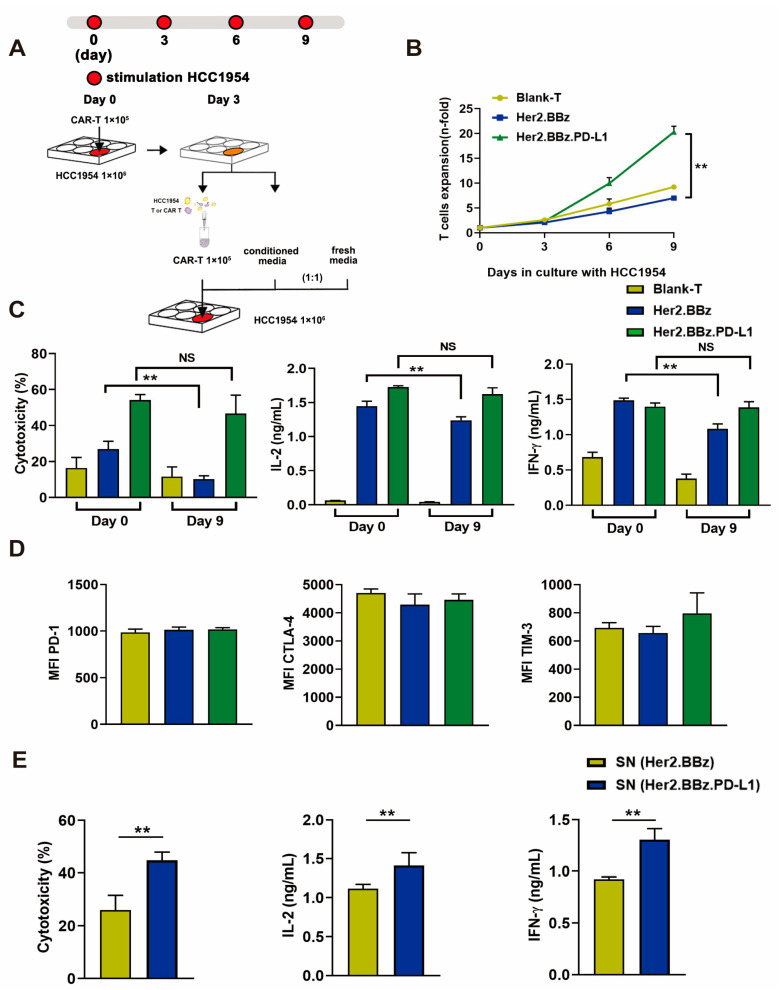
Effects of antigenic stimulation on cell exhaustion, cytotoxicity, cell proliferation, and cytokine secretion by bland T, Her2.BBz CAR T, and Her2.BBz.PD-L1 CAR T cells. (**A**) Scheme of the experimental procedure: 1 × 10^5^ blank T cells and CAR T cells were co-cultured with 1 × 10^6^ HCC1954 cells for 3 days. The culture supernatant (conditioned media) was collected and filtered. T cells were isolated by MACS CD3 microbeads, and 1 × 10^5^ T cells resuspended in a medium containing equal amounts of conditioned medium and fresh medium, and co-cultured with 1 × 10^6^ HCC1954 cells for another 3 days. This process was repeated one more time at day 6, and the cell exhaustion experiment lasted for 9 days in total. (**B**) During the prolonged antigenic stimulation, T cells were isolated by CD3 microbeads and analyzed for cell proliferation (*n* = 3). (**C**) The cytotoxicity and the cytokine secretion were measured before (day 0) and after (day 9) antigen stimulation at the E:T ratio of 1:10 (*n* = 3). (**D**) The expression of T cell surface markers PD-1, CTLA-4, TIM-3 was measured by flow cytometry, and MFI was calculated using FlowJo (*n* = 3). (**E**) The effects of PD-L1 secreted by CAR T cells on the cytotoxicity and cytokine secretion by Her2.BBz CAR T cells. Her2.BBz CAR T cells were co-cultured with HCC1954-Luc cells for 48 h at the E:T ratio of 10:1 (*n* = 3), SN: supernatant. A total of 4 healthy donors were enrolled in our studies and we present data from a representative individual. One-way ANOVA was performed to determine statistical significance in (**B**–**D**), *t* test was performed to determine statistical significance in (**E**). The results are means ± SD (** *p* < 0.05, compared with Her2.BBz CAR T cells group; NS, no significance).

**Figure 5 ijms-24-04197-f005:**
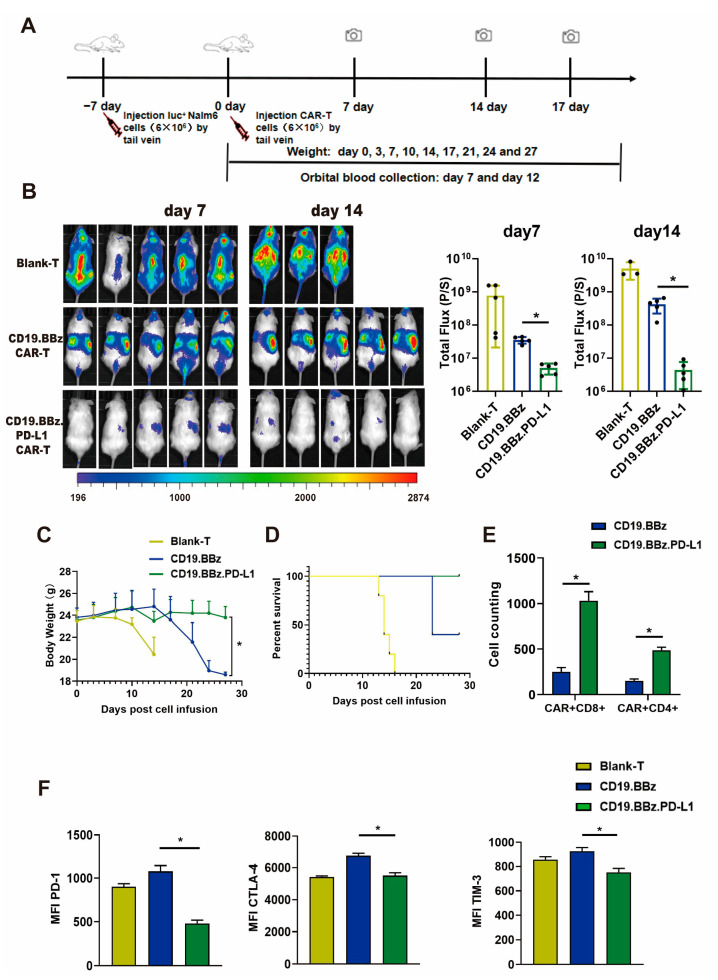
Anti-tumor effect of blank T, CD19.BBz CAR T and CD19.BBz.PD-L1 CAR T cells against Nalm6-Luc xenograft tumors. (**A**) Scheme of the experimental procedure. (**B**) Tumor burden was analyzed by bioluminescent imaging at day 7 and 14 (scattered plot, *n* = 5). (**C**) Mouse weight curves (*n* = 5). (**D**) Kaplan–Meier curve of survival. (**E**) T cells were isolated from 0.5 mL blood by CD3 microbeads and the number the CAR T cells was counted and the amount of CD4^+^ and CD8^+^ cells was analyzed by flow cytometry (*n* = 3). (**F**) The expression of the T cell surface markers PD-1, CTLA-4, and TIM-3 was measured by flow cytometry, and MFI was calculated using FlowJo (*n* = 3). A total of three healthy donors were enrolled in in vivo experiments and the mice experiment was performed once for each donor. We present data from a representative individual. One-way ANOVA analysis was performed to determine statistical significance, the results are means ± SD. (* *p* < 0.05, compared with CD19.BBz CAR T cells group).

**Figure 6 ijms-24-04197-f006:**
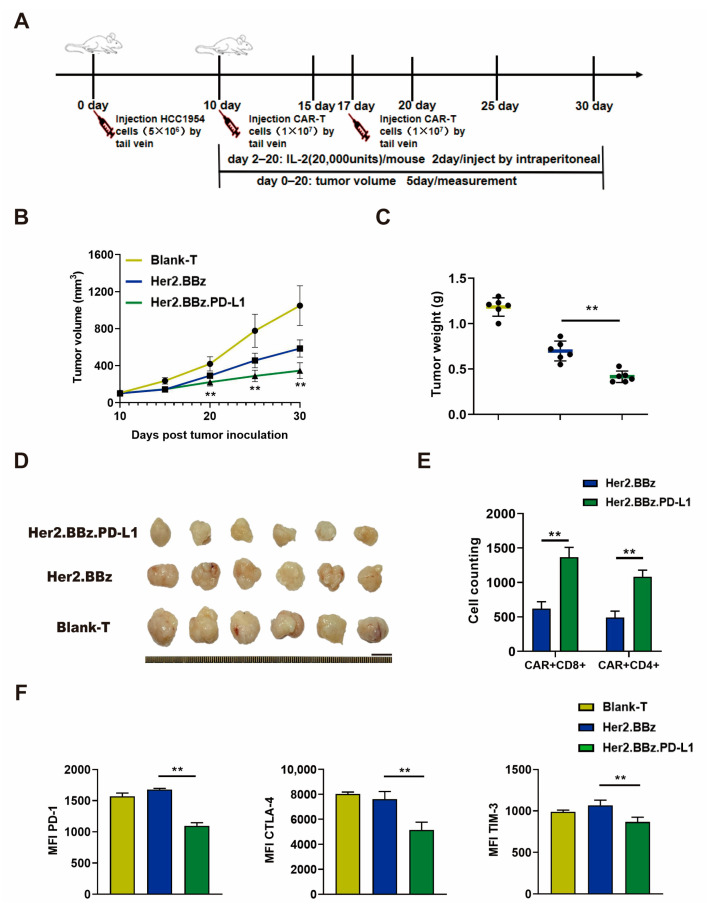
Anti-tumor effect of blank T, Her2.BBz CAR T, and Her2.BBz.PD-L1 CAR T cells on HCC1954 xenograft tumors. (**A**) Schematic illustration of the experimental procedure. (**B***–***D**) Tumor growth curves (*n* = 6) (**B**), tumor weights (scattered plot, *n* = 6) (**C**), and tumor images (**D**) of HCC1954 xenograft tumors in NCG mice treated with CAR T cells. (**E**) T cells were isolated from 0.2 g tumor by CD3 microbeads, the number the CAR T cells was counted, and the amount of CD4^+^ and CD8^+^ cells was analyzed by flow cytometry (*n* = 3). (**F**) The expression of surface markers PD-1, CTLA-4, and TIM-3 in tumor infiltrating T cells was measured by flow cytometry, and MFI was calculated using FlowJo (*n* = 3). A total of three healthy donors were enrolled in in vivo experiments and the mice experiment was performed once for each donor. We present data from a representative individual. One-way ANOVA analysis was performed to determine statistical significance, the results are means ± SD (** *p* < 0.05, compared with Her2.BBz CAR T cells group).

## Data Availability

Not applicable.

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
