# Peer review of "4-1BB-Based CAR T Cells Effectively Reverse Exhaustion and Enhance the Anti-Tumor Immune Response through Autocrine PD-L1 scFv Antibody"

_ijms, 2023, doi:10.3390/ijms24044197_

Round 1

Reviewer 1 Report

Authors have described the efficacy of 4-1BB CAR T with autocrine PD-L1 scFv antibody activity exhibiting better antitumor outcomes than what is currently available. It is a well designed study and properly written manuscript. Whiles the in vivo data is quite convincing, it would also be interesting to see what the images from the other 2 mice not shown in figure 5B look like. There were 5 mice per treatment and 3 were shown in figure 5B.

Reviewer 2 Report

Revision of the review : “4-1BB based CAR T cells effectively reverse exhaustion and enhance the anti-tumor immune response through autocrine PD-L1 scFv antibody”.

The article describes the association of CARs with the co-stimulatory 41BB domain against liquid (CD19 positive) or solid tumor ( HER2 positive) with a secreted PD-L1 scFv antibody. The authors compare their engineered CAR with scFv PD-L1 with the conventional CARs and show that the CAR with PD-L1 have a better response against the tumor and in the in vivo model there is a decrease in T cells exhaustion markers.

The article is interesting for the scientific community however there are several revisions need to be addressed before its publication.

Revisions needed to be addressed:

In this study, it would be important to compare the effect of PD-L1 scFv secreted by the T cells in comparison to administrated PD-L1 antibody.

It would be interesting to compare the phenotype of the conventional CAR with the CAR associated with the secreted scFv PD-L1 to better understand the results obtained.  

To better understand whether the T cell engineered with the CAR and the scFv PD-L1 improves exhaustion, as claimed, I would suggest that the authors perform an experiment in which the cells are exposed to the target cells in different rounds, i.e. will get several stimulations. Thus allowing to better address the T cells exhaustion status.

Line 14: Claiming that the exhaustion of CAR-T cells is the key limitation, I do not agree with that statement. There are other crucial limitation such as tumor infiltration and the tumor microenvironment.

Line 33: Can you please provide a ref? Can you please indicate that CD19-CAR T cells is already used for the treatment of leukemia with a high successful rate. 

In Results: Can you please quantify the amount of PD-L1 scFv secreted by the CAR?

Line 86: From what I understand the vector is composed by pCDH-EF1a-"CAR"-CMV-copGFP, is it correct? Please clarify.

Line 94: Can you please demonstrate that the expression of GFP is proportional to the expression of the CAR? For that purpose, you can use a CD19-Fc to detect the CD19 CAR and an anti-scFv to detect HER2 CAR and compare with the GFP expression in the T cells.

Line 110: I could not find the figures Figure S1A and Figure S1B.

Line 113: Please indicate how the cytotoxicity was determined. Can you please include the controls of Nalm6 without T cells and the "100%" killing activity?

Figure 1: Please indicate in the figure the localization of the GFP in the vector.

Figure 1B: Please correct in the legend of the figure B (flow cytometry), to copGFP ( or GFP) and not CAR-FITC. I do not agree in using CAR-FITC, as what it is detected is the GFP expression and not the CAR, moreover, it has not been shown that the expression of the CAR is correlated with GFP.

Figure 1C: Can you please show the flow cytometry for the expression of PD-L1 in those cells to have a more quantitative and comparable data.

Figure 2: Please indicate the statistics in comparison to the T cells for both CAR as it is important to validate the activity of the CAR against the target cells.

Considering that the CAR activity against the target cells should be higher than the T cells, please indicate why it is not the case in 2:1 E:T ratio in Nalm6 and more significantly for the HER2 CAR vs T cells.

Please indicate/clarify how many donors were used to isolate the T cells? Is this an experiment in which the T cells were obtained from only one donor? If yes, did you engineered other T cells from different donors and observed similar results? Same comment for figure 3 and 4.

Figure 3: Please clarify the Figure 3A with the procedure as it is not easily understandable!

Figure 3E: Please indicate how many days of exposure to the target and the ratio E:T.

Figure 4A: Same comment as for figure 3A.

Figure 4E: Please indicate the time and ratio E:T

Figure 5: Please indicate in figure A, the type of injection used for each procedure, it will help the reader to better follow the experiment.

Please indicate the number of mice used and how many time this experiment was repeated.

Figure 6: Please indicate the type of injection used for each procedure, it will help the reader to better follow the experiment.

Line 130: Please comment the high non-specific killing observed for the T cells, which is similar to the CAR.

Line 133: Can you please indicate in the material and methods how those cells were obtained?

Line 135: Same comments as for Nalm6-luc cytotoxicity activity determination and control!

Line 136: Can you please show (even if in supplementary data) the results obtained for cytokine secretion at the other effector to target ratios?

Line 178: results 2.5: Can you please explain why the CAR HER2 is not more active/cytotoxic against the target cells when compared with the T cells alone? Although the production of IL2 and IFNg is higher???`

Line 284: Can you please comment the efficiency in killing the tumor?

Line 314: Please indicate the REF.

Line 356: Please indicate which sequences/ clone of the scFv were used for CD19 and HER2 CARs.

Line 369: Please briefly describe how the lentivirus were obtained and its titres.

Line 380: You mean transduced? Please correct, as well as in the below text.

Line 382: at which temperature?

Line 404: Please indicate: Phosphate Buffered Saline for PBS. Please be careful with the nomenclature, abbreviations should be use once define in the text. Verify it in the text.

Line 408: Please indicate the temperature!

Line 411: HCC1954 are adherent cells, please indicate how they were prepared for the immunostaining.

Line 414: Please indicate how those cells were generated/obtained?

Line 404: Please clarify the unit mpk i.e. mg/kg/day

Round 2

Reviewer 2 Report

Revision of the review : “Half-life extension and biodistribution modulation of biotherapeutics via red blood cell hitch-hiking with novel anti-band 3 single-domain antibodies”.

The authors replied to all the question and doubts, however it is still clear that major revisions are still required as mentioned below.

Some revisions needed to be done:

-       Please correct the supplementary figure. Make two supplementary figures, separating the figure S1C, as it is not related with the other figures A and B.

-       Please include as supplementary figure the results obtained for FACS shown that the expression of GFP is proportional to the CAR (your response 9)

-       Please indicate in the respective figures that the results obtained are from one donor and it is representative of four donor (this information should be present in material and methods, but as well in the legends).

-       Figure 3A is now better explain in the material and methods. However the scheme is still hard to follow, please change it as well as, explain better the procedure in the figure legend. Same comment for 4A.

-       Please indicate in the figure that the mice experiment was performed only once. This information should be clear in the results. Considering that it was done only once, how can you be sure of the reproducibility of the results?

-       Several of your responses claim that the difference observed may be due to the proportion of the population of CD4/CD8 T cells. Do you have data that support such hypothesis?

-       Taking in consideration your response 28, please indicate in that the observation in cytokine release was observed only after 3 days of culture and comment on it in the article. For the figure 2A, can you as well indicated after how many days of co-culture the cytokines were measured.

Round 3

Reviewer 2 Report

Revision of the review : “Half-life extension and biodistribution modulation of biotherapeutics via red blood cell hitch-hiking with novel anti-band 3 single-domain antibodies”.

I appreciated the effort that the authors have done to reply to the question and doubts. The last responses to my questions and the correction done in the article, improve it significantly and now I consider that it is acceptable for publication.

Please do a last revision of the paper as there are still minor text correction.

I would as well recommend to put the data of “response 7” in supplementary data to support those claims in the article.
